# Robust Bi-Tempered Logistic Loss
# Based on Bregman Divergences

**Ehsan Amid** [⋆†]     **Manfred K. Warmuth** [⋆†]     **Rohan Anil** [†]     **Tomer Koren** [†§]

⋆ Department of Computer Science, University of California, Santa Cruz
§ School of Computer Science, Tel Aviv University, Tel Aviv, Israel
† Google Brain
{eamid,manfred,rohananil,tkoren}@google.com

## Abstract

We introduce a temperature into the exponential function and replace the softmax output layer of the neural networks by a high-temperature generalization. Similarly, the logarithm in the loss we use for training is replaced by a low-temperature logarithm. By tuning the two temperatures, we create loss functions that are non-convex already in the single layer case. When replacing the last layer of the neural networks by our bi-temperature generalization of the logistic loss, the training becomes more robust to noise. We visualize the effect of tuning the two temperatures in a simple setting and show the efficacy of our method on large datasets. Our methodology is based on Bregman divergences and is superior to a related two-temperature method that uses the Tsallis divergence.

## 1   Introduction

The logistic loss, also known as the softmax loss, has been the standard choice in training deep neural networks for classification. The loss involves the application of the softmax function on the activations of the last layer to form the class probabilities followed by the relative entropy (aka the Kullback-Leibler (KL) divergence) between the true labels and the predicted probabilities. The logistic loss is known to be a convex function of the activations (and consequently, the weights) of the last layer.

Although desirable from an optimization standpoint, convex losses have been shown to be prone to outliers [15] as the loss of each individual example unboundedly increases as a function of the activations. These outliers may correspond to extreme examples that lead to large gradients, or misclassified training examples that are located far away from the classification boundary. Requiring a convex loss function at the output layer thus seems somewhat arbitrary, in particular since convexity in the last layer's activations does not guarantee convexity with respect to the parameters of the network outside the last layer. Another issue arises due to the exponentially decaying tail of the softmax function that assigns probabilities to the classes. In the presence of mislabeled training examples near the classification boundary, the short tail of the softmax probabilities enforces the classifier to stretch the decision boundary towards the noisy training examples. In contrast, heavy-tailed alternatives for the softmax probabilities have been shown to significantly improve the robustness of the loss to these examples [8].

The logistic loss is essentially the negative logarithm of the predicted class probabilities, which are computed as the normalized exponentials of the inputs. In this paper, we tackle both shortcomings of the logistic loss, pertaining to its convexity as well as its tail-lightness, by replacing the logarithm and the exponential functions with their corresponding "tempered" versions. We define the function

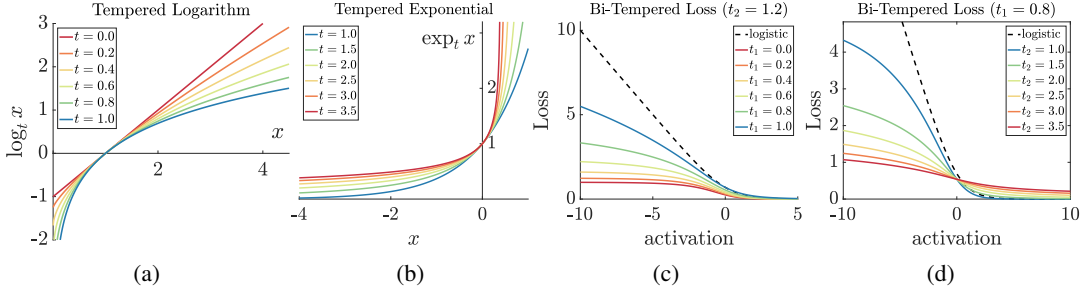

Figure 1: Tempered logarithm and exponential functions, and the bi-tempered logistic loss: (a) $\log_t$ function, (b) $\exp_t$ function, bi-tempered logistic loss when (c) $t_2 = 1.2$ fixed and $t_1 \leqslant 1$, and (d) $t_1 = 0.8$ fixed and $t_2 \geqslant 1$.

$\log_t : \mathbb{R}_+ \to \mathbb{R}$ with *temperature* parameter $t \geqslant 0$ as in [16]:

$$\log_t(x) \coloneqq \frac{1}{1-t}(x^{1-t} - 1). \tag{1}$$

The $\log_t$ function is monotonically increasing and concave. The standard (natural) logarithm is recovered at the limit $t \to 1$. Unlike the standard log, the $\log_t$ function is bounded from below by $-1/(1-t)$ for $0 \leqslant t < 1$. This property will be used to define bounded loss functions that are significantly more robust to outliers. Similarly, our heavy-tailed alternative for the softmax function is based on the tempered exponential function. The function $\exp_t : \mathbb{R} \to \mathbb{R}_+$ with temperature $t \geqslant 0$ is defined as the inverse[1] of $\log_t$, that is,

$$\exp_t(x) \coloneqq [1 + (1-t)\,x]_+^{1/(1-t)} \,, \tag{2}$$

where $[\,\cdot\,]_+ = \max\{\,\cdot\,, 0\}$. The standard exp function is again recovered at the limit $t \to 1$. Compared to the exp function, a heavier tail (for negative values of $x$) is achieved for $t > 1$. We use this property to define heavy-tailed analogues of softmax probabilities at the output layer.

The vanilla logistic loss can be viewed as a logarithmic (relative entropy) divergence that operates on a "matching" exponential (softmax) probability assignment [11, 12]. Its convexity then stems from classical convex duality, using the fact that the probability assignment function is the gradient of the dual function to the negative entropy on the simplex. When the $\log_{t_1}$ and $\exp_{t_2}$ are substituted instead, this duality still holds whenever $t_1 = t_2$, albeit with a different Bregman divergence, and the induced loss remains convex[2]. However, for $t_1 < t_2$, the loss becomes non-convex in the output activations. In particular, $0 \leqslant t_1 < 1$ leads to a bounded loss, while $t_2 > 1$ provides tail-heaviness. Figure 1 illustrates the tempered $\log_t$ and $\exp_t$ functions as well as examples of our proposed bi-tempered logistic loss function for a two-class problem expressed as a function of the activation of the first class. The true label is assumed to be class one.

Tempered generalizations of the logistic regression have been introduced before [7, 8, 22, 2]. The most recent two-temperature method [2] is based on the Tsallis divergence and contains all the previous methods as special cases. However, the Tsallis based divergences do not result in proper loss functions. In contrast, we show that the Bregman based construction introduced in this paper is indeed proper, which is a requirement for many real-world applications.

## 1.1 Our replacement of the softmax output layer in neural networks

Consider an arbitrary classification model with multiclass softmax output. We are given training examples of the form $(\boldsymbol{x}, \boldsymbol{y})$, where $\boldsymbol{x}$ is a fixed dimensional input vector and the target $\boldsymbol{y}$ is a probability vector over $k$ classes. In practice, the targets are often one-hot encoded binary vectors in $k$ dimensions. Each input $\boldsymbol{x}$ is fed to the model, resulting in a vector $\boldsymbol{z}$ of inputs to the final softmax layer. This layer typically has one trainable weight vector $\boldsymbol{w}_i$ per class $i$ and yields the predicted class probability

$$\hat{y}_i = \frac{\exp(\hat{a}_i)}{\sum_{j=1}^{k} \exp(\hat{a}_j)} = \exp\left(\hat{a}_i - \log\sum_{j=1}^{k}\exp(\hat{a}_j)\right), \text{ for linear activation } \hat{a}_i = \boldsymbol{w}_i \cdot \boldsymbol{z} \text{ for class } i.$$

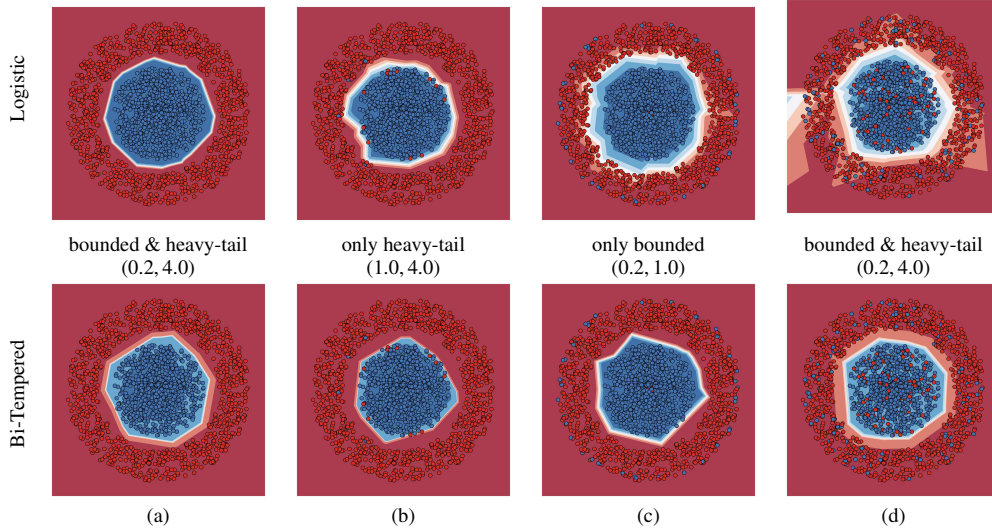

Figure 2: Logistic vs. robust bi-tempered logistic loss: (a) noise-free labels, (b) small-margin label noise, (c) large-margin label noise, and (d) random label noise. The temperature values $(t_1, t_2)$ for the bi-tempered loss are shown above each figure.

We first replace the softmax function by a generalized heavy-tailed version that uses the $\exp_{t_2}$ function with $t_2 > 1$, which we call the *tempered softmax function*:

$$\hat{y}_i = \exp_{t_2}\left(\hat{a}_i - \lambda_{t_2}(\hat{\boldsymbol{a}})\right), \quad \text{where} \quad \lambda_{t_2}(\hat{\boldsymbol{a}}) \in \mathbb{R} \ \text{is s.t.} \ \sum_{j=1}^{k} \exp_{t_2}\left(\hat{a}_j - \lambda_{t_2}(\hat{\boldsymbol{a}})\right) = 1 \,.$$

This requires computing the normalization value $\lambda_{t_2}(\hat{\boldsymbol{a}})$ (for each example) via a binary search or an iterative procedure like the one given in Appendix A. The relative entropy between the true label $\boldsymbol{y}$ and prediction $\hat{\boldsymbol{y}}$ is replaced by the tempered version with temperature range $0 \leqslant t_1 < 1$,

$$\sum_{i=1}^{k}\left(y_i\left(\log_{t_1} y_i - \log_{t_1}\hat{y}_i\right) - \tfrac{1}{2-t_1}\left(y_i^{2-t_1} - \hat{y}_i^{2-t_1}\right)\right) \overset{\text{if } \boldsymbol{y} \text{ one-hot}}{=} -\log_{t_1}\hat{y}_c - \tfrac{1}{2-t_1}\left(1 - \sum_{i=1}^{k}\hat{y}_i^{2-t_1}\right)\,.$$

where $c = \operatorname{argmax}_i y_i$ is the index of the one-hot class. In later sections we prove various properties of this loss. When $t_1 = t_2 = 1$, then it reduces to the vanilla relative entropy loss with softmax. Also when $0 \leqslant t_1 < 1$, then the loss is bounded, while $t_2 > 1$ gives the tempered softmax function a heavier tail.

## 1.2 An illustration

We provide some intuition on why both boundedness of the loss as well as tail-heaviness of the tempered softmax are crucial for robustness. For this, we train a small two-layer feed-forward neural network on a synthetic binary classification problem in two dimensions. The network has 10 and 5 units in the first and second layer, respectively[3]. Figure 2(a) shows the results of the logistic and our bi-tempered logistic loss on the noise-free dataset. The network converges to a desirable classification boundary (the white stripe in the figure) using both loss functions. In Figure 2(b), we illustrate the effect of adding small-margin label noise to the training examples, targeting those examples that reside near the noise-free classification boundary. The logistic loss clearly follows the noisy examples by stretching the classification boundary. On the other hand, using *only* the tail-heavy tempered softmax function ($t_2 = 4$ while $t_1 = 1$, i.e. KL divergence as the divergence) can handle the noisy examples by producing more uniform class probabilities. Next, we show the effect of large-margin noisy examples in Figure 2(c), targeting examples that are located far away from the noise-free classification boundary. The convexity of the logistic loss causes the network to be highly affected by the noisy examples that are located far away from the boundary. In contrast, *only* the boundedness of the loss ($t_1 = 0.2$ while $t_2 = 1$, meaning that the outputs are vanilla softmax probabilities) reduces the

effect of the outliers by allocating at most a finite amount of loss to each example. Finally, we show the effect of random label noise that includes both small-margin and large-margin noisy examples in Figure 2(d). Clearly, the logistic loss fails to handle the noise, while our bi-tempered logistic loss successfully recovers the appropriate boundary. Note for random noise, we exploit *both* boundedness of the loss ($t_1 = 0.2 < 1$) as well as the tail-heaviness of the probability assignments ($t_2 = 4 > 1$).

The theoretical background as well as our treatment of the softmax layer of the neural networks are developed in later sections. In particular, we show that special discrete choices of the temperatures result in a large variety of divergences commonly used in machine learning. As we show in our experiments, tuning the two temperatures as continuous parameters is crucial.

## 1.3 Summary of the experiments

We perform experiments by adding synthetic label noise to MNIST and CIFAR-100 datasets and compare the results of our robust bi-tempered loss to the vanilla logistic loss. Our bi-tempered loss is significantly more robust to label noise (when trained on noisy data and test accuracy is measured w.r.t. the clean data): It provides 98.56% and 62.55% accuracy on MNIST and CIFAR-100, respectively, when trained with 40% label noise (compared to 97.64% and 53.17%, respectively, obtained using logistic loss). The bi-tempered loss also yields improvement over the state-of-the-art results on the ImageNet-2012 dataset using both the Resnet18 and Resnet50 architectures (see Table 2).

## 2 Preliminaries

### 2.1 Convex duality and Bregman divergences on the simplex

We start by briefly reviewing some basic background in convex analysis. For a continuously-differentiable strictly convex function $F : \mathcal{D} \to \mathbb{R}$, with convex domain $\mathcal{D} \subseteq \mathbb{R}^k$, the Bregman divergence [3] between $\boldsymbol{y}, \hat{\boldsymbol{y}} \in \mathcal{D}$ induced by $F$ is defined as

$$\Delta_F(\boldsymbol{y}, \hat{\boldsymbol{y}}) = F(\boldsymbol{y}) - F(\hat{\boldsymbol{y}}) - (\boldsymbol{y} - \hat{\boldsymbol{y}}) \cdot f(\hat{\boldsymbol{y}}) \,,$$

where $f(\hat{\boldsymbol{y}}) := \nabla F(\hat{\boldsymbol{y}})$ denotes the gradient of $F$ at $\hat{\boldsymbol{y}}$ (sometimes called the link function of $F$). Clearly $\Delta_F(\boldsymbol{y}, \hat{\boldsymbol{y}}) \geqslant 0$ and $\Delta_F(\boldsymbol{y}, \hat{\boldsymbol{y}}) = 0$ iff $\boldsymbol{y} = \hat{\boldsymbol{y}}$. Also the Bregman divergence is always convex in the first argument and $\nabla_{\boldsymbol{y}} \Delta_F(\boldsymbol{y}, \hat{\boldsymbol{y}}) = f(\boldsymbol{y}) - f(\hat{\boldsymbol{y}})$, but not generally in its second argument. Bregman divergence generalizes many well-known divergences such as the squared Euclidean $\Delta_F(\boldsymbol{y}, \hat{\boldsymbol{y}}) = \frac{1}{2} \|\boldsymbol{y} - \hat{\boldsymbol{y}}\|_2^2$ (with $F(\boldsymbol{y}) = \frac{1}{2} \|\boldsymbol{y}\|_2^2$) and the Kullback–Leibler divergence $\Delta_F(\boldsymbol{y}, \hat{\boldsymbol{y}}) = \sum_i (y_i \log \frac{y_i}{\hat{y}_i} - y_i + \hat{y}_i)$ (with $F(\boldsymbol{y}) = \sum_i (y_i \log y_i - y_i)$). The Bregman divergence is typically not symmetric, i.e. $\Delta_F(\boldsymbol{y}, \hat{\boldsymbol{y}}) \neq \Delta_F(\hat{\boldsymbol{y}}, \boldsymbol{y})$. Additionally, the Bregman divergence is invariant to adding affine functions to the convex function $F$: $\Delta_{F+A}(\boldsymbol{y}, \hat{\boldsymbol{y}}) = \Delta_F(\boldsymbol{y}, \hat{\boldsymbol{y}})$, where $A(\boldsymbol{y}) = b + \boldsymbol{c} \cdot \boldsymbol{y}$ for arbitrary $b \in \mathbb{R}$, $\boldsymbol{c} \in \mathbb{R}^k$.

For every differentiable strictly convex function $F$ (with domain $\mathcal{D} \subseteq \mathbb{R}_+^k$), there exists a convex dual $F^* : \mathcal{D}^* \to \mathbb{R}$ function such that for dual parameter pairs $(\boldsymbol{y}, \boldsymbol{a})$, $\boldsymbol{a} \in \mathcal{D}^*$, the following holds: $\boldsymbol{a} = f(\boldsymbol{y})$ and $\boldsymbol{y} = f^*(\boldsymbol{a}) = \nabla F^*(\boldsymbol{a}) = f^{-1}(\boldsymbol{a})$. However, we are mainly interested in the dual of the function $F$ when the domain is restricted to the probability simplex $S^k := \{\boldsymbol{y} \in \mathbb{R}_+^k \mid \sum_{i=1}^k y_i = 1\}$. Let $\check{F}^* : \check{\mathcal{D}}^* \to \mathbb{R}$ denote the convex conjugate of the restricted function $F : \mathcal{D} \cap S^k \to \mathbb{R}$,

$$\check{F}^*(\boldsymbol{a}) = \sup_{\boldsymbol{y}' \in \mathcal{D} \cap S^k} \left( \boldsymbol{y}' \cdot \boldsymbol{a} - F(\boldsymbol{y}') \right) = \sup_{\boldsymbol{y}' \in \mathcal{D}} \inf_{\lambda \in \mathbb{R}} \left( \boldsymbol{y}' \cdot \boldsymbol{a} - F(\boldsymbol{y}') + \lambda \left( 1 - \sum_{i=1}^k y_i' \right) \right) ,$$

where we introduced a Lagrange multiplier $\lambda \in \mathbb{R}$ to enforce the linear constraint $\sum_{i=1}^k y_i' = 1$. At the optimum, the following relationships hold between the primal and dual variables:

$$f(\boldsymbol{y}) = \boldsymbol{a} - \lambda(\boldsymbol{a}) \, \boldsymbol{1} \quad \text{and} \quad \boldsymbol{y} = f^{-1} \left( \boldsymbol{a} - \lambda(\boldsymbol{a}) \, \boldsymbol{1} \right) = \check{f}^*(\boldsymbol{a}) , \tag{3}$$

where $\lambda(\boldsymbol{a})$ is chosen so that it satisfies the constraint. Note the dependence of the optimum $\lambda$ on $\boldsymbol{a}$.

### 2.2 Matching losses

Next, we recall the notion of a *matching loss* [11, 12, 4, 17]. It arises as a natural way of defining a loss function over activations $\hat{\boldsymbol{a}} \in \mathbb{R}^k$, by first mapping them to a probability distribution over class labels using a *transfer function* $s : \mathbb{R}^k \to S^k$, and then computing a *divergence* $\Delta_F$ between this

distribution and the correct target labels. The idea behind the following definition is to "match" the transfer function and the divergence via duality.[4]

**Definition 1** (Matching Loss). *Let $F : S^k \to \mathbb{R}$ be a continuously-differentiable, strictly convex function and let $s : \mathbb{R}^k \to S^k$ be a transfer function such that $\hat{y} = s(\hat{a})$ denotes the predicted probability distribution based on the activations $\hat{a}$. Then the loss function*

$$L_F(\hat{a} \mid y) := \Delta_F(y, s(\hat{a})),$$

*is called the matching loss for $s$, if $s = \check{f}^* = \nabla \check{F}^*$.*

Note that $\check{f}^*$ is no longer one-to-one since $\check{f}^*(\hat{a} + \mathbb{R} \, \mathbf{1}) = \check{f}^*(\hat{a})$ (see Appendix D for more details). However, w.l.o.g. we can constrain the domain of the function to $\hat{a} \in \mathbf{dom}(\check{f}^*) \cap \{a' \in \mathbb{R}^k \mid a' \cdot \mathbf{1} = 0\}$ to obtain a one-to-one mapping. The matching loss is useful due to the following property.

**Proposition 1.** *The matching loss $L_F(\hat{a} \mid y)$ is convex w.r.t. the activations $\hat{a} \in \mathbf{dom}(\check{f}^*) \cap \{a' \in \mathbb{R}^k \mid a' \cdot \mathbf{1} = 0\}$.*

*Proof.* Note that $\check{F}^*$ is a strictly convex function and the following relation holds between the divergences induced by $F$ and $\check{F}^*$ (see proof of Proposition 4 in Appendix D):

$$\Delta_F(y, \hat{y}) = \Delta_{\check{F}*}\big((\check{f}^*)^{-1}(\hat{y}), \, (\check{f}^*)^{-1}(y)\big). \tag{4}$$

Thus for any $\hat{a}$ in the range of $(\check{f}^*)^{-1}$,

$$\Delta_F\big(y, \check{f}^*(\hat{a})\big) = \Delta_{\check{F}*}\big(\hat{a}, (\check{f}^*)^{-1}(y)\big).$$

The claim now follows from the convexity of $\Delta_{\check{F}*}$ w.r.t. its first argument. $\qquad\square$

The original motivating example for the matching loss was the logistic loss [11, 12]. It can be obtained as the matching loss for the softmax function

$$\hat{y}_i = [\check{f}^*(\hat{a})]_i = \frac{\exp(\hat{a}_i)}{\sum_{j=1}^{k} \exp(\hat{a}_j)},$$

which corresponds to the relative entropy (KL) divergence

$$L_F(\hat{a} \mid y) = \Delta_F\big(y, \check{f}^*(\hat{a})\big) = \sum_{i=1}^{k} y_i \, (\log y_i - \log \hat{y}_i) = \sum_{i=1}^{k} \big(y_i \, \log y_i - y_i \, \hat{a}_i\big) + \log \Big(\sum_{i=1}^{k} \exp(\hat{a}_i)\Big),$$

induced from the negative entropy function $F(y) = \sum_{i=1}^{k}(y_i \log y_i - y_i)$. We next define a family of convex functions $F_t$ parameterized by a temperature $t \geqslant 0$. The matching loss $L_{F_t}(\hat{a} \mid y) = \Delta_{F_t}\big(y, \check{f}_t^*(\hat{a})\big)$ for the link function $\check{f}_t^*$ of $\check{F}_t^*$ is convex in the activations $\hat{a}$. However, by letting the temperature $t_2$ of $\check{f}_{t_2}^*$ be larger than the temperature $t_1$ of $F_{t_1}$, we construct bounded non-convex losses with heavy-tailed transfer functions.

## 3 Tempered Matching Loss

We start by introducing a generalization of the relative entropy divergence, denoted by $\Delta_{F_t}$, induced by a strictly convex function $F_t : \mathbb{R}_+^k \to \mathbb{R}$ with a temperature parameter $t \geqslant 0$. The convex function $F_t$ is chosen so that its gradient takes the form[5] $f_t(y) := \nabla F_t(y) = \log_t y$. Via simple integration, we obtain that

$$F_t(y) = \sum_{i=1}^{k} \Big(y_i \log_t y_i + \tfrac{1}{2-t}(1 - y_i^{2-t})\Big).$$

Indeed, $F_t$ is a convex function since $\nabla^2 F_t(y) = \mathrm{diag}(y^{-t}) \geq 0$ for any $y \in \mathbb{R}_+^k$. In fact, $F_t$ is strongly convex, for $0 \leqslant t \leqslant 1$:

**Lemma 1.** *The function $F_t$, with $0 \leqslant t \leqslant 1$, is $B^{-t}$–strongly convex over the set $\{y \in \mathbb{R}_+^k : \|y\|_{2-t} \leqslant B\}$ w.r.t. the $L_{2-t}$-norm.*

See Appendix B for a proof. The Bregman divergence induced by $F_t$ is then given by

$$
\begin{aligned}
\Delta_{F_t}(\boldsymbol{y}, \hat{\boldsymbol{y}}) &= \sum_{i=1}^{k} \left( y_i \log_t y_i - y_i \log_t \hat{y}_i - \tfrac{1}{2-t} y_i^{2-t} + \tfrac{1}{2-t} \hat{y}_i^{2-t} \right) \\
&= \sum_{i=1}^{k} \left( \tfrac{1}{(1-t)(2-t)} y_i^{2-t} - \tfrac{1}{1-t} y_i \hat{y}_i^{1-t} + \tfrac{1}{2-t} \hat{y}_i^{2-t} \right).
\end{aligned}
\tag{5}
$$

The second form may be recognized as $\beta$-divergence [5] with parameter $\beta = 2 - t$. The divergence (5) includes many well-known divergences such as squared Euclidean, KL, and Itakura-Saito divergence as special cases. A list of additional special cases is given in Table 3 of Appendix C.

The following corollary is the direct consequence of the strong convexity of $F_t$.

**Corollary 1.** *Let* $\max(\|\boldsymbol{y}\|_{2-t}, \|\hat{\boldsymbol{y}}\|_{2-t}) \leqslant B$ *for* $0 \leqslant t < 1$. *Then*

$$
\frac{1}{2B^t} \|\boldsymbol{y} - \hat{\boldsymbol{y}}\|_{2-t}^2 \leqslant \Delta_{F_t}(\boldsymbol{y}, \hat{\boldsymbol{y}}) \leqslant \frac{B^t}{2(1-t)^2} \|\boldsymbol{y}^{1-t} - \hat{\boldsymbol{y}}^{1-t}\|_{\frac{2-t}{1-t}}^2 .
$$

See Appendix B for a proof. Thus for $0 \leqslant t < 1$, $\Delta_{F_t}(\boldsymbol{y}, \hat{\boldsymbol{y}})$ is upper-bounded by $\frac{2B^{2-t}}{(1-t)^2}$. Note that boundedness on the simplex also implies boundedness in the $L_{2-t}$-ball. Thus, Corollary 1 immediately implies the boundedness of the divergence $\Delta_{F_t}(\boldsymbol{y}, \hat{\boldsymbol{y}})$ with $0 \leqslant t < 1$ over the simplex. Alternate parameterizations of the family $\{F_t\}$ of convex functions and their corresponding Bregman divergences are discussed in Appendix C.

### 3.1 Tempered softmax function

Now, let us consider the convex function $F_t(\boldsymbol{y})$ when its domain is restricted to the probability simplex $S^k$. We denote the constrained dual of $F_t(\boldsymbol{y})$ by $\check{F}_t^*(\boldsymbol{a})$,

$$
\check{F}_t^*(\boldsymbol{a}) = \sup_{\boldsymbol{y}' \in S^k} \left( \boldsymbol{y}' \cdot \boldsymbol{a} - F_t(\boldsymbol{y}') \right) = \sup_{\boldsymbol{y}' \in \mathbb{R}_+^k} \inf_{\lambda_t \in \mathbb{R}} \left( \boldsymbol{y}' \cdot \boldsymbol{a} - F_t(\boldsymbol{y}') + \lambda_t \left( 1 - \sum_{i=1}^{k} y_i' \right) \right).
\tag{6}
$$

Following our discussion in Section 2.1 and using (3), the transfer function induced by $\check{F}_t^*$ is[6]

$$
\boldsymbol{y} = \exp_t \left( \boldsymbol{a} - \lambda_t(\boldsymbol{a}) \mathbf{1} \right), \quad \text{with } \lambda_t(\boldsymbol{a}) \text{ s.t. } \sum_{i=1}^{k} \exp_t \left( a_i - \lambda_t(\boldsymbol{a}) \right) = 1.
\tag{7}
$$

### 3.2 Matching loss of tempered softmax

Finally, we derive the matching loss function $L_{F_t}$. Plugging in (7) into (5), we have

$$
L_t(\hat{\boldsymbol{a}} \mid \boldsymbol{y}) = \Delta_{F_t} \left( \boldsymbol{y}, \exp_t(\hat{\boldsymbol{a}} - \lambda_t(\hat{\boldsymbol{a}}) \mathbf{1}) \right).
$$

Recall that by Proposition 1, this loss is convex in activations $\hat{\boldsymbol{a}} \in \mathbf{dom}(\check{f}^*) \cap \{\boldsymbol{a}' \in \mathbb{R}^k \mid \boldsymbol{a}' \cdot \mathbf{1} = 0\}$. In general, $\lambda_t(\boldsymbol{a})$ does not have a closed form solution. However, it can be easily approximated via an iterative method, e.g., a binary search. An alternative (fixed-point) algorithm for computing $\lambda_t(\boldsymbol{a})$ for $t > 1$ is given in Algorithm 1 of Appendix A.

## 4 Robust Bi-Tempered Logistic Loss

A more interesting class of loss functions can be obtained by introducing a "mismatch" between the temperature of the divergence function (5) and the temperature of the probability assignment function, i.e. the tempered softmax (7). That is, we consider loss functions of the following type:

$$
\boxed{\forall \, 0 \leqslant t_1 < 1 < t_2 : L_{t_1}^{t_2}(\hat{\boldsymbol{a}} \mid \boldsymbol{y}) := \Delta_{F_{t_1}} \left( \boldsymbol{y}, \exp_{t_2}(\hat{\boldsymbol{a}} - \lambda_{t_2}(\hat{\boldsymbol{a}}) \mathbf{1}) \right), \text{with } \lambda_t(\hat{\boldsymbol{a}}) \text{ s.t.} \sum_{i=1}^{k} \exp_t \left( a_i - \lambda_t(\boldsymbol{a}) \right) = 1.}
\tag{8}
$$

We call this the *Bi-Tempered Logistic Loss*. As illustrated in our two-dimensional example in Section 1, both properties are crucial for handling noisy examples. The derivative of the bi-tempered loss is given in Appendix E. In the following, we discuss the properties of this loss for classification.

## 4.1 Properness and Monte-Carlo sampling

Let $P_{UK}(x, y)$ denote the (unknown) joint probability distribution of the observed variable $x \in \mathbb{R}^m$ and the class label $y \in [k]$. The goal of discriminative learning is to approximate the posterior distribution of the labels $P_{UK}(y \mid x)$ via a parametric model $P(y \mid x; \Theta)$ parameterized by $\Theta$. Thus the model fitting can be expressed as minimizing the following expected loss between the data and the model's label probabilities

$$\mathbb{E}_{P_{UK}(x)}\left[ \Delta\big(P_{UK}(y \mid x), P(y \mid x; \Theta)\big) \right], \tag{9}$$

where $\Delta\big(P_{UK}(y \mid x), P(y \mid x; \Theta)\big)$ is any divergence measure between $P_{UK}(y \mid x)$ and $P(y \mid x; \Theta)$. We use $\Delta := \Delta_{F_{t_1}}$ as the divergence and $P(i \mid x; \Theta) := P(y = i \mid x; \Theta) = \exp_{t_2}(\hat{a}_i - \lambda_{t_2}(\hat{a}))$, where $\hat{a}$ is the activation vector of the last layer given input $x$ and $\Theta$ is the set of all weights of the network. Ignoring the constant terms w.r.t. $\Theta$, our loss (9) becomes

$$\mathbb{E}_{P_{UK}(x)}\left[ \sum_i \Big( -P_{UK}(i \mid x)\log_t P(i \mid x; \Theta) + \frac{1}{2-t} P(i \mid x; \Theta)^{2-t}\Big) \right] \tag{10a}$$

$$= -\mathbb{E}_{P_{UK}(x,y)}\left[ \log_t P(y \mid x; \Theta) \right] + \mathbb{E}_{P_{UK}(x)}\left[ \frac{1}{2-t}\sum_i P(i \mid x; \Theta)^{2-t}\Big) \right] \tag{10b}$$

$$\approx \frac{1}{N}\sum_n \Big( -\log_t P(y_n \mid x_n; \Theta) + \frac{1}{2-t}\sum_i P(i \mid x_n; \Theta)^{2-t} \Big), \tag{10c}$$

where from (10b) to (10c), we perform a Monte-Carlo approximation of the expectation w.r.t. $P_{UK}(x, y)$ using samples $\{(x_n, y_n)\}_{n=1}^N$. Thus, (10c) is an unbiased approximate of the expected loss (9), thus is a *proper* loss [20].

Following the same approximation steps for the Tsallis divergence used in [2], we have

$$\mathbb{E}_{P_{UK}(x)}\Bigg[ \underbrace{-\sum_i P_{UK}(i \mid x)\log_t \frac{P(i \mid x; \Theta)}{P_{UK}(i \mid x)}}_{\Delta_t^{\text{Tsallis}}\big(P_{UK}(y|x), P(y|x;\Theta)\big)} \Bigg] \approx -\frac{1}{N}\sum_n \log_t \frac{P(y_n \mid x_n; \Theta)}{P_{UK}(y_n \mid x_n)},$$

which, due to the fact that $\log_t \frac{a}{b} \neq \log_t a - \log_t b$ in general, requires access to the (unknown) conditional distribution $P_{UK}(y \mid x)$. In this case the approximation $-\frac{1}{N}\sum_n \log_t P(y_n \mid x_n; \Theta)$ proposed in [2] by setting $P_{UK}(y_n \mid x_n)$ to 1 is not an unbiased estimator of (9) and therefore, not proper.

## 4.2 Bayes-risk consistency

Another important property of a multiclass loss is the Bayes-risk consistency [19]. Bayes-risk consistency of the two-temperature logistic loss based on the Tsallis divergence was shown in [2]. As expected, the tempered Bregman loss (8) is also Bayes-risk consistent even in the non-convex case.

**Proposition 2.** *The multiclass bi-tempered logistic loss $L_{t_1}^{t_2}(\hat{a} \mid y)$ is Bayes-risk consistent.*

## 5 Experiments

We demonstrate the practical utility of the bi-tempered logistic loss function on a wide variety of image classification tasks. For moderate-size experiments, we use MNIST dataset of handwritten digits [14] and CIFAR-100, which contains real-world images from 100 different classes [13]. We use ImageNet-2012 [6] for large scale image classification, having 1000 classes. All experiments are carried out using the TensorFlow [1] framework. We use P100 GPU's for small-scale experiments and Cloud TPU-v2 for larger scale ImageNet-2012 experiments. An implementation of the bi-tempered logistic loss is available online at: `https://github.com/google/bi-tempered-loss`.

### 5.1 Corrupted labels experiments

For our moderate size datasets, i.e. MNIST and CIFAR-100, we introduce noise by artificially corrupting a fraction of the labels and producing a new set of labels for each noise level. For all experiments, we compare our bi-tempered loss function against the logistic loss.

| Dataset | Loss | Label Noise Level | | | | | |
|---------|------|------|------|------|------|------|------|
| | | 0.0 | 0.1 | 0.2 | 0.3 | 0.4 | 0.5 |
| MNIST | Logistic | **99.40** | 98.96 | 98.70 | 98.50 | 97.64 | 96.13 |
| | Bi-Tempered (0.5, 4.0) | 99.24 | **99.13** | **99.02** | **98.62** | **98.56** | **97.69** |
| CIFAR-100 | Logistic | 74.03 | 69.94 | 66.39 | 63.00 | 53.17 | 52.96 |
| | Bi-Tempered (0.8, 1.2) | **75.30** | **73.30** | **70.69** | **67.45** | **62.55** | **57.80** |

Table 1: Top-1 accuracy on a clean test set for MNIST and CIFAR-100 datasets where a fraction of the training labels are corrupted.

For MNIST, we use a CNN with two convolutional layers of size 32 and 64 with a mask size of 5, followed by two fully-connected layers of size 1024 and 10. We apply max-pooling after each convolutional layer with a window size equal to 2 and use dropout during training with keep probability equal to 0.75. We use the AdaDelta optimizer [21] with 500 epochs and batch size of 128 for training.

| Model | Logistic | Bi-tempered (0.9,1.05) |
|-------|----------|------------------------|
| Resnet18 | $71.333 \pm 0.069$ | **71.618** $\pm 0.163$ |
| Resnet50 | $76.332 \pm 0.105$ | **76.748** $\pm 0.164$ |

Table 2: Top-1 accuracy on ImageNet-2012 with Resnet-18 and 50 architectures.

For CIFAR-100, we use a Resnet-56 [10] model without batch norm from [9] with SGD + momentum optimizer trained for 50k steps with batch size of 128 and use the standard learning rate stair case decay schedule. For both experiments, we report the test accuracy of the checkpoint which yields the highest accuracy on an identically label-noise corrupted validation set. We search over a set of learning rates for each experiment. For both experiments, we exhaustively search over a number of temperatures within the range $[0.5, 1)$ and $(1.0, 4.0]$ for $t_1$ and $t_2$, respectively. The results are presented in Table 1 where we report the top-1 accuracy on a clean test set. As can be seen, the bi-tempered loss outperforms the logistic loss for all noise levels (including the noise-free case for CIFAR-100). Using our bi-tempered loss function the model is able to continue to perform well even for high levels of label noise whereas the accuracy of the logistic loss drops immediately with a much smaller level of noise.

## 5.2 Large scale experiments

We train state-of-the-art Resnet-18 and Resnet-50 models on the ImageNet-2012 dataset. Note that the ImageNet-2012 dataset is inherently noisy due to some amount of mislabeling. We train on a 4x4 CloudTPU-v2 device with a batch size of 4096. All experiments were trained for 180 epochs, and use the SGD + momentum optimizer with staircase learning rate decay schedule. The results are presented in Table 2. For both architectures we see a significant gain in the top-1 accuracy using the robust bi-tempered loss.

## 6 Conclusion and Future Work

Neural networks on large standard datasets have been optimized along with a large variety of variables such as architecture, transfer function, choice of optimizer, and label smoothing to name just a few. We proposed a new variant by training the network with tunable loss functions. We do this by first developing convex loss functions based on temperature dependent logarithm and exponential functions. When both temperatures are the same, then a construction based on the notion of "matching loss" leads to loss functions that are convex in the last layer. However by letting the temperature of the new tempered softmax function be larger than the temperature of the tempered log function used in the divergence, we construct tunable losses that are non-convex in the last layer. Our construction remedies two issues simultaneously: we construct bounded tempered loss functions that can handle large-margin outliers and introduce heavy-tailedness in our new tempered softmax function that seems to handle small-margin mislabeled examples. At this point, we simply took a number of benchmark datasets and networks for these datasets that have been heavily optimized for the logistic loss paired with vanilla softmax and simply replaced the loss in the last layer by our new construction. By simply trying a number of temperature pairs, we already achieved significant improvements. We believe that with a systematic "joint optimization" of all commonly tried variables, significant further improvements can be achieved. This is of course a more long-term goal. We also plan to explore the idea of annealing the temperature parameters over the training process.

## Acknowledgement

We would like to thank Jerome Rony for pointing out that early stopping improves the accuracy of the logistic loss on the noisy MNIST experiment. This research was partially supported by the NSF grant IIS-1546459.

## Footnotes

[1] When $0 \leqslant t < 1$, the domain of $\exp_t$ needs to be restricted to $-1/(1-t) \leqslant x$ for the inverse property to hold.

[2] In a restricted domain when $t_1 = t_2 < 1$, as discussed later.

[3]An interactive visualization of the bi-tempered loss is available at: `https://google.github.io/bi-tempered-loss/`

[4]Originally in [11, 12], the matching loss was defined as a simple integral over the transfer function $s = f^{-1}$: $L_F(\hat{a} \mid y) = \int_{s^{-1}(y)}^{\hat{a}} (s(z) - y) \cdot d\,z$. Our new duality based definition handles additional linear constraints.

[5]Here, the $\log_t$ function is applied elementwise.

[6]Note that due to the simplex constraint, the link function $\boldsymbol{y} = \check{f}_t^*(\boldsymbol{a}) = \nabla \check{F}_t^*(\boldsymbol{a}) = \exp_t \left( \boldsymbol{a} - \lambda_t(\boldsymbol{a}) \mathbf{1} \right)$ is different from $f_t^{-1}(\boldsymbol{a}) = f_t^*(\boldsymbol{a}) = \nabla F_t^*(\boldsymbol{a}) = \exp_t(\boldsymbol{a})$, i.e., the gradient of the unconstrained dual.

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
