[Supplementary Material]

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

129   This matching is useful due to the following.

130   **Proposition 1.** *The matching loss $L_F(\hat{\boldsymbol{a}} \mid \boldsymbol{y})$ is always convex w.r.t. the activations $\hat{\boldsymbol{a}}$.*

131   *Proof.* Note that $\check{F}^*$ is a strictly convex function and the following relation holds between the
132   divergences induced by $F$ and $\check{F}^*$:

$$\Delta_F(\boldsymbol{y}, \hat{\boldsymbol{y}}) = \Delta_{\check{F}*}\big((\check{f}^*)^{-1}(\hat{\boldsymbol{y}}), \, (\check{f}^*)^{-1}(\boldsymbol{y})\big). \tag{4}$$

133   Thus for any $\hat{\boldsymbol{a}}$ in the range of $(\check{f}^*)^{-1}$,

$$\Delta_F\big(\boldsymbol{y}, \check{f}^*(\hat{\boldsymbol{a}})\big) = \Delta_{\check{F}*}\big(\hat{\boldsymbol{a}}, (\check{f}^*)^{-1}(\boldsymbol{y})\big).$$

134   The claim now follows from the convexity of the Bregman divergence $\Delta_{\check{F}*}$ w.r.t. its first argument.
135   $\qquad\qquad\qquad\qquad\qquad\qquad\qquad\qquad\qquad\qquad\qquad\qquad\qquad\qquad\qquad\qquad\qquad$ $\square$

136   The original motivating example for the matching loss was the logistic loss [9, 10]. It can be obtained
137   as the matching loss for the softmax function

$$\hat{y}_i = [\check{f}^*(\hat{\boldsymbol{a}})]_i = \frac{\exp(\hat{a}_i)}{\sum_{j=1}^{k} \exp(\hat{a}_j)},$$

138   which corresponds to the relative entropy (KL) divergence

$$L_F(\hat{\boldsymbol{a}} \mid \boldsymbol{y}) = \Delta_F\big(\boldsymbol{y}, \check{f}^*(\hat{\boldsymbol{a}})\big) = \sum_{i=1}^{k} y_i \,(\log y_i - \log \hat{y}_i) = \sum_{i=1}^{k} \big(y_i \, \log y_i - y_i \, \hat{a}_i\big) + \log \Big(\sum_{i=1}^{k} \exp(\hat{a}_i)\Big),$$

139   induced from the negative entropy function $F(\boldsymbol{y}) = \sum_{i=1}^{k}(y_i \log y_i - y_i)$. We next define a family
140   of convex functions $F_t$ parameterized by a temperature $t \geqslant 0$. The matching loss $L_{F_t}(\hat{\boldsymbol{a}} \mid \boldsymbol{y}) = $
141   $\Delta_{F_t}\big(\boldsymbol{y}, \check{f}_t^*(\hat{\boldsymbol{a}})\big)$ for the link function $\check{f}_t^*$ of $\check{F}_t^*$ is always convex in the activations $\hat{\boldsymbol{a}}$. However, by letting
142   the temperature $t_2$ of $\check{f}_{t_2}^*$ be larger than the temperature $t_1$ of $F_{t_1}$, we construct bounded non-convex
143   losses with heavy-tailed transfer functions.

# 3   Tempered Bregman divergence

145   We start by introducing a generalization of the relative entropy, denoted by $\Delta_{F_t}$, induced by a strictly
146   convex function $F_t : \mathbb{R}_+^k \to \mathbb{R}$ with a temperature parameter $t \geqslant 0$. The convex function $F_t$ is chosen
147   so as its gradient takes the form[3] $f_t(\boldsymbol{y}) := \nabla F_t(\boldsymbol{y}) = \log_t \boldsymbol{y}$. Via simple integration, we obtain that

$$F_t(\boldsymbol{y}) = \sum_{i=1}^{k} \big(y_i \log_t y_i + \tfrac{1}{2-t}(1 - y_i^{2-t})\big).$$

148   Indeed, $F_t$ is a convex function since $\nabla^2 F_t(\boldsymbol{y}) = \mathrm{diag}(\boldsymbol{y}^{-t}) \succeq 0$ for any $\boldsymbol{y} \in \mathbb{R}_+^k$. In fact, $F_t$ is strongly
149   convex, for $0 \leqslant t \leqslant 1$:

150   **Lemma 1.** *The function $F_t$, with $0 \leqslant t \leqslant 1$, is $B^{-t}$–strongly convex over the set $\{\boldsymbol{y} \in \mathbb{R}_+^k : \|\boldsymbol{y}\|_{2-t} \leqslant B\}$*
151   *w.r.t. the $L_{2-t}$-norm.*

152   See Appendix B for a proof. The Bregman divergence induced by $F_t$ is then given by

$$\begin{aligned}
\Delta_{F_t}(\boldsymbol{y}, \hat{\boldsymbol{y}}) &= \sum_{i=1}^{k} \big(y_i \log_t y_i - y_i \log_t \hat{y}_i - \tfrac{1}{2-t}y_i^{2-t} + \tfrac{1}{2-t}\hat{y}_i^{2-t}\big) \\
&= \sum_{i=1}^{k} \Big(\tfrac{1}{(1-t)(2-t)} y_i^{2-t} - \tfrac{1}{1-t} y_i \hat{y}_i^{1-t} + \tfrac{1}{2-t} \hat{y}_i^{2-t}\Big).
\end{aligned} \tag{5}$$

153   The second form may be recognized as $\beta$-divergence [4] with parameter $\beta = 2 - t$. The divergence (5)
154   includes many well-known divergences such as squared Euclidean, KL, and Itakura-Saito divergence
155   as special cases. A list of additional special cases is given in Table 3 of Appendix C.

156   The following corollary is the direct consequence of the strong convexity of $F_t$, for $0 \leqslant t < 1$.

157 **Corollary 1.** *Let* $\max(\|\mathbf{y}\|_{2-t}, \|\hat{\mathbf{y}}\|_{2-t}) \leqslant B$ *for* $0 \leqslant t < 1$. *Then*

$$\frac{1}{2B^t}\|\mathbf{y} - \hat{\mathbf{y}}\|_{2-t}^2 \leqslant \Delta_{F_t}(\mathbf{y}, \hat{\mathbf{y}}) \leqslant \frac{B^t}{2(1-t)^2}\|\mathbf{y} - \hat{\mathbf{y}}\|_{2-t}^{2(1-t)}.$$

158 See Appendix B for a proof. Thus for $0 \leqslant t < 1$, $\Delta_{F_t}(\mathbf{y}, \hat{\mathbf{y}})$ is upper-bounded by $\frac{B^{2-t}}{(1-t)^2}$. Note that
159 boundedness on the simplex also implies boundedness in the $L_{2-t}$-ball of radius 1. Thus, Corollary 1
160 immediately implies the boundedness of the divergence $\Delta_{F_t}(\mathbf{y}, \hat{\mathbf{y}})$ with $0 \leqslant t < 1$ over the simplex.
161 Alternate parameterizations of the family $\{F_t\}$ of convex functions and their corresponding Bregman
162 divergences are discussed in Appendix C.

### 3.1 Tempered transfer function

164 Now, let us consider the convex function $F_t(\mathbf{y})$ when its domain is restricted to the probability simplex
165 $S^k$. We denote the constrained dual of $F_t(\mathbf{y})$ by $\check{F}_t^*(\mathbf{a})$,

$$\check{F}_t^*(\mathbf{a}) = \sup_{\mathbf{y}' \in S^k} \left(\mathbf{y}' \cdot \mathbf{a} - F_t(\mathbf{y}')\right) = \sup_{\mathbf{y}' \in \mathbb{R}_+^k} \inf_{\lambda_t \in \mathbb{R}} \left(\mathbf{y}' \cdot \mathbf{a} - F_t(\mathbf{y}') + \lambda_t \left(1 - \sum_{i=1}^k y_i'\right)\right). \quad (6)$$

166 Following our discussion in Section 2.1 and using (3), the transfer function induced by $\check{F}_t^*$ is[4]

$$\mathbf{y} = \exp_t\left(\mathbf{a} - \lambda_t(\mathbf{a})\,\mathbf{1}\right), \quad \text{with } \lambda_t(\mathbf{a}) \text{ s.t. } \sum_{i=1}^k \exp_t\left(a_i - \lambda_t(\mathbf{a})\right) = 1. \quad (7)$$

### 3.2 Tempered matching losses

Finally, we derive the matching loss function $L_{F_t}$. Plugging in (7) into (5), we have

$$L_t(\hat{\mathbf{a}} \mid \mathbf{y}) = \Delta_{F_t}\left(\mathbf{y}, \exp_t(\hat{\mathbf{a}} - \lambda_t(\hat{\mathbf{a}}))\right).$$

168 Recall that by Proposition 1, this loss is convex in activations $\hat{\mathbf{a}}$. In general, $\lambda_t(\mathbf{a})$ does not have a
169 closed form solution. However, it can be easily approximated via an iterative method, e.g., a binary
170 search. An alternative (fixed-point) algorithm for computing $\lambda_t(\mathbf{a})$ for $t > 1$ is given in Algorithm 1
171 of Appendix A.

## 4  Robust Bi-Tempered Logistic Loss

173 A more interesting class of loss functions can be obtained by introducing a "mismatch" between
174 the temperature of the divergence function (5) and the temperature of the probability assignment
175 function (7). That is, we consider loss functions of the following type:

$$\forall\, 0 \leqslant t_1 < 1 < t_2:\; L_{t_1}^{t_2}(\hat{\mathbf{a}} \mid \mathbf{y}) := \Delta_{F_{t_1}}\left(\mathbf{y}, \exp_{t_2}(\hat{\mathbf{a}} - \lambda_{t_2}(\hat{\mathbf{a}}))\right), \text{ with } \lambda_t(\hat{\mathbf{a}}) \text{ s.t. } \sum_{i=1}^k \exp_t\left(a_i - \lambda_t(\mathbf{a})\right) = 1. \quad (8)$$

176 We call this the *Bi-Tempered Logistic Loss*. Note that for the prescribed range of the two temperatures,
177 the loss is bounded and has a heavier-tailed probability assignment function compared to the vanilla
178 softmax function. As illustrated in our 2-dimensional example in Section 1, both properties are
179 crucial for handling noisy examples. The derivative of the bi-tempered loss are given in Appendix E.
180 In the following, we discuss the properties of this loss for classification.

### 4.1 Properness and Monte-Carlo sampling

182 Let $P_{\text{UK}}(\mathbf{x}, y)$ denote the (unknown) joint probability distribution of the observed variable $\mathbf{x} \in \mathbb{R}^m$ and
183 the class label $y \in [k]$. The goal of discriminative learning is to approximate the posterior distribution
184 of the labels $P_{\text{UK}}(y \mid \mathbf{x})$ via a parametric model $P(y \mid \mathbf{x}; \Theta)$ parameterized by $\Theta$. Thus the model fitting

can be expressed as minimizing the following expected loss between the data and the model label posterior probabilities

$$\mathbb{E}_{P_{\text{UK}}(\boldsymbol{x})}\Big[\Delta\big(P_{\text{UK}}(y\mid\boldsymbol{x}),P(y\mid\boldsymbol{x};\Theta)\big)\Big]\,, \tag{9}$$

where $\Delta\big(P_{\text{UK}}(y\mid\boldsymbol{x}),P(y\mid\boldsymbol{x};\Theta)\big)$ is any proper divergence measure between $P_{\text{UK}}(y\mid\boldsymbol{x})$ and $P(y\mid\boldsymbol{x};\Theta)$. We use $\Delta := \Delta_{F_{t_1}}$ as the divergence and $P(y = i\mid\boldsymbol{x};\Theta) := P(i\mid\boldsymbol{x};\Theta) = \exp_{t_2}(\hat{a}_i - \lambda_{t_2}(\hat{\boldsymbol{a}}))$, where $\hat{\boldsymbol{a}}$ is the activation vector of the last layer given input $\boldsymbol{x}$ and $\Theta$ is the set of all weights of the network. Ignoring the constant terms w.r.t. $\Theta$, our loss (9) becomes

$$\mathbb{E}_{P_{\text{UK}}(\boldsymbol{x})}\Big[\sum_i\big(-P_{\text{UK}}(i\mid\boldsymbol{x})\log_t P(i\mid\boldsymbol{x};\Theta) + \frac{1}{2-t}P(i\mid\boldsymbol{x};\Theta)^{2-t}\big)\Big] \tag{10a}$$

$$\approx \frac{1}{N}\sum_n\sum_i\big(-P_{\text{UK}}(i\mid\boldsymbol{x}_n)\log_t P(i\mid\boldsymbol{x}_n;\Theta) + \frac{1}{2-t}P(i\mid\boldsymbol{x}_n;\Theta)^{2-t}\big) \tag{10b}$$

$$\approx \frac{1}{N}\sum_n\big(-\log_t P(y_n\mid\boldsymbol{x}_n;\Theta) + \sum_i\frac{1}{2-t}P(i\mid\boldsymbol{x}_n;\Theta)^{2-t}\big)\,, \tag{10c}$$

where from (10a) to (10b), we perform a Monte-Carlo approximation of the expectation w.r.t. $P_{\text{UK}}(\boldsymbol{x})$ using samples $\{\boldsymbol{x}_n\}_{n=1}^N$ and in (10c), we approximate the expectation w.r.t. each $P_{\text{UK}}(i\mid\boldsymbol{x}_n)$ using a single sample $y_n$. Thus, (10c) is an unbiased approximate of the expected loss (9), thus is a *proper* loss [18].

Following the same approximation steps for the Tsallis divergence, we have

$$\mathbb{E}_{P_{\text{UK}}(\boldsymbol{x})}\Big[\underbrace{-\sum_i P_{\text{UK}}(i\mid\boldsymbol{x})\log_t\frac{P(i\mid\boldsymbol{x};\Theta)}{P_{\text{UK}}(i\mid\boldsymbol{x})}}_{\Delta_t^{\text{Tsallis}}\big(P_{\text{UK}}(y\mid\boldsymbol{x}),P(y\mid\boldsymbol{x};\Theta)\big)}\Big] \approx -\frac{1}{N}\sum_n\log_t\frac{P(y_n\mid\boldsymbol{x}_n;\Theta)}{P_{\text{UK}}(y_n\mid\boldsymbol{x}_n)}\,,$$

which, due to the fact that $\log_t\frac{a}{b} \neq \log_t a - \log_t b$ in general, requires access to the label posterior distribution $P_{\text{UK}}(y\mid\boldsymbol{x})$. Thus, the approximation $-\frac{1}{N}\sum_n\log_t P(y_n\mid\boldsymbol{x}_n;\Theta)$ proposed in [2] by approximating $P_{\text{UK}}(y_n\mid\boldsymbol{x}_n)$ by 1 is not an unbiased estimator of (9) and therefore, not proper.

## 4.2 Bayes-risk consistency

Another important property of a multiclass loss is the Bayes-risk consistency [17]. Bayes-risk consistency of the two-temperature logistic loss based on the Tsallis divergence was shown in [2]. As expected, the tempered Bregman loss (8) is also Bayes-risk consistent, even in the non-convex case.

**Proposition 2.** *The multiclass bi-tempered logistic loss $L_{t_1}^{t_2}(\hat{\boldsymbol{a}}\mid y)$ is Bayes-risk consistent.*

## 5 Experiments

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

[2]In a restricted domain when $t_1 = t_2 < 1$, as discussed later.

[3]Here, the $\log_t$ function is applied elementwise.

[4]Note that due to the simplex constraint, the link function $\mathbf{y} = \check{f}_t^*(\mathbf{a}) = \nabla \check{F}_t^*(\mathbf{a}) = \exp_t\left(\mathbf{a} - \lambda_t(\mathbf{a})\right)$ is different from $f_t^{-1}(\mathbf{a}) = f^*(\mathbf{a}) = \nabla F_t^*(\mathbf{a}) = \exp_t(\mathbf{a})$, i.e., the gradient of the unconstrained dual.

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

## A An Iterative Algorithm for Computing the Normalization

---

**Algorithm 1** Iterative algorithm for computing $\lambda_t(\boldsymbol{a})$ (from [2])

---

    **Input:** Vector of activations $\boldsymbol{a}$, temperature $t > 1$
    $\mu \leftarrow \max(\boldsymbol{a})$
    $\tilde{\boldsymbol{a}} \leftarrow \boldsymbol{a} - \mu$
    **while** $\tilde{a}$ not converged **do**
        $Z(\tilde{\boldsymbol{a}}) \leftarrow \sum_{i=1}^{k} \exp_t(\tilde{a}_i)$
        $\tilde{\boldsymbol{a}} \leftarrow Z(\tilde{\boldsymbol{a}})^{1-t}(\boldsymbol{a} - \mu \mathbf{1})$
    **end while**
    **Return:** $\lambda_t(\boldsymbol{a}) \leftarrow -\log_t \frac{1}{Z(\tilde{\boldsymbol{a}})} + \mu$

---

## B Strong Convexity and Smoothness

The following material for strong convexity and strong smoothness are adopted from [16].

**Definition 2** ($\sigma$-Strong Convexity). A continuous function $F$ is $\sigma$-strongly convex w.r.t. the norm $\|\cdot\|$ over the convex set $\mathcal{S}$ if $\mathcal{S}$ is contained in the domain of $F$ and for any $\boldsymbol{u}, \boldsymbol{v} \in \mathcal{S}$, we have

$$F(\boldsymbol{v}) \geqslant F(\boldsymbol{u}) + \nabla F(\boldsymbol{u}) \cdot (\boldsymbol{v} - \boldsymbol{u}) + \frac{\sigma}{2} \|\boldsymbol{v} - \boldsymbol{u}\|^2 \,.$$

**Lemma 2.** *Assume $F$ is twice differentiable. Then $F$ is $\sigma$-strongly convex if*
$$\left(\nabla^2 F(\boldsymbol{u})\, \boldsymbol{v}\right) \cdot \boldsymbol{v} \geqslant \sigma \|\boldsymbol{v}\|^2), \quad \forall \boldsymbol{u}, \boldsymbol{v} \in \mathcal{S} \,.$$

**Lemma 3.** *Let $F$ be a $\sigma$-strongly convex function over the non-empty convex set $\mathcal{S}$. For all $\boldsymbol{u}, \boldsymbol{v} \in \mathcal{S}$, we have*
$$\frac{\sigma}{2} \|\boldsymbol{u} - \boldsymbol{v}\|^2 \leqslant \Delta_F(\boldsymbol{v}, \boldsymbol{u}) \,.$$

***Proof of Lemma 1.*** We have $\nabla^2 F(\boldsymbol{u}) = \operatorname{diag}(\boldsymbol{u}^{-t})$. Applying Lemma 3, note that the function

$$(\nabla^2 F_t(\boldsymbol{u}) \cdot \boldsymbol{v}) \cdot \boldsymbol{v} = \sum_i \frac{v_i^2}{u_i^t} \,,$$

is unbounded over the set $\mathcal{S} = \{\boldsymbol{v} \in \mathbb{R}_+^d : \|\boldsymbol{v}\|_{2-t} \leqslant B\}$ and the minimum happens at the boundary $\{\|\boldsymbol{v}\|_{2-t} = B\}$.

$$\min_{\boldsymbol{v}} \sum_i \frac{v_i^2}{u_i^t} + \gamma(\sum_i v_i^{2-t} - 1) \;\Rightarrow\; \boldsymbol{v} = B \, \frac{\boldsymbol{u}}{\|\boldsymbol{u}\|_{2-t}} \,,$$

where $\gamma$ is the Lagrange multiplier. Plugging in the solution yields $\sum_i \frac{v_i^2}{u_i^t} \geqslant \frac{1}{B^t} \|\boldsymbol{v}\|_{2-t}^2$. $\qquad\square$

**Definition 3** ($\sigma$-Strong Smoothness). A function differentiable function $G$ is $\sigma$-strongly smooth w.r.t. the norm $\|\cdot\|$ if
$$\Delta_G(\boldsymbol{v}, \boldsymbol{u}) \leqslant \frac{\sigma}{2} \|\boldsymbol{v} - \boldsymbol{u}\|^2 \,.$$

**Lemma 4.** *Let $F$ be a closed and convex function. Then $F$ is $\sigma$-strongly convex w.r.t. the $|\cdot\|$ if and only if $F^*$, the dual of $F$, is $\frac{1}{\sigma}$-strongly smooth w.r.t. the dual norm $\|\cdot\|_*$.*

***Proof of Corollary 1.*** Note that using the duality of the Bregman divergences, we have
$$\Delta_{F_t}(\boldsymbol{y}, \hat{\boldsymbol{y}}) = \Delta_{F_t^*}(f_t(\hat{\boldsymbol{y}}), f_t(\boldsymbol{y})) = \Delta_{F_t^*}(\log_t(\hat{\boldsymbol{y}}), \log_t(\boldsymbol{y})) \,.$$

Using the strong convexity of $F_t$ and strong smoothness of $F_t^*$, we have
$$\frac{1}{2B^t} \|\boldsymbol{y} - \hat{\boldsymbol{y}}\|_{2-t}^2 \leqslant \Delta_{F_t}(\boldsymbol{y}, \hat{\boldsymbol{y}}) \leqslant \frac{B^t}{2} \| \log_t \boldsymbol{y} - \log_t \hat{\boldsymbol{y}}\|_{\frac{2-t}{1-t}}^2 \,.$$

Note that $\|\cdot\|_{2-t}$ and $\|\cdot\|_{\frac{2-t}{1-t}}$ are dual norms. Substituting the definition of $\log_t$ to the right-hand-side, we have
$$\| \log_t \boldsymbol{y} - \log_t \hat{\boldsymbol{y}}\|_{2-t}^2 = \frac{1}{2(1-t)^2} \|\boldsymbol{y}^{1-t} - \hat{\boldsymbol{y}}^{1-t}\|_{\frac{2-t}{1-t}}^2 = \|\boldsymbol{y} - \hat{\boldsymbol{y}}\|_{2-t}^{2(1-t)} \,.$$

$\qquad\square$

# C  Other Tempered Convex Functions

We begin with a list of interesting temperature choices for the convex function $F_t$ and its induced divergence:

| $t$ | $F_t(\boldsymbol{y})$ | $\Delta_{F_t}(\boldsymbol{y},\hat{\boldsymbol{y}})$ | Name |
|---|---|---|---|
| $0$ | $\frac{1}{2}\|\boldsymbol{y}\|_2^2$ | $\frac{1}{2}\|\boldsymbol{y}-\hat{\boldsymbol{y}}\|_2^2$ | Euclidean |
| $\frac{1}{2}$ | $\frac{1}{3}\sum_i(4y_i^{\frac{4}{3}}-6y_i+2)$ | $\sum_i(\frac{4}{3}y_i^{\frac{3}{2}}-2y_i\sqrt{\hat{y}_i}+\frac{3}{2}\hat{y}_i^{\frac{3}{2}})$ | |
| $1$ | $\sum_i(y_i\log y_i - y_i + 1)$ | $\sum_i(y_i\log\frac{y_i}{\hat{y}_i}-y_i+\hat{y}_i)$ | KL-divergence |
| $\frac{3}{2}$ | $\sum_i(-4y_i^{\frac{3}{2}}+2y_i+2)$ | $2\sum_i\frac{(\sqrt{y_i}-\sqrt{\hat{y}_i})^2}{\sqrt{\hat{y}_i}}$ | Squared Xi on roots |
| $2$ | $\sum_i(-\log y_i + y_i)$ | $\sum_i(\frac{y_i}{\hat{y}_i}-\log\frac{y_i}{\hat{y}_i}-1)$ | Itakura-Saito |
| $3$ | $\frac{1}{2}\sum_i(-\frac{1}{y_i}+y_i-2)$ | $\frac{1}{2}\sum_i(\frac{1}{y_i}-\frac{2}{\hat{y}_i}+\frac{y_i}{\hat{y}_i^2})$ | Inverse |

Table 3: Some special cases of the tempered Bregman divergence.

In the construction of the convex function family $F_t$ we used $F_t(x) = \int \log_t(x)$ exploiting the fact that $\log_t(x)$ is strictly increasing. We can also define an alternative convex function family $\widetilde{F}_t$ by utilizing the convexity (respectively, concavity) of the $\log_t$ function for values of $t \geqslant 0$ (respectively, $t \leqslant 0$):

$$\widetilde{F}_t(\boldsymbol{y}) = -\frac{1}{t}\sum_i(\log_t y_i - y_i + 1) = -\frac{1}{t(1-t)}\sum_i(y_i^{1-t}-y_i).$$

Note that $\widetilde{f}_t(\boldsymbol{y}) := \nabla\widetilde{F}(\boldsymbol{y}) = \frac{1-y^{-t}}{t}$ and $\nabla^2\widetilde{F}_t(\boldsymbol{y}) = \mathrm{diag}(\boldsymbol{y}^{-(1+t)})$, thus $\widetilde{F}_t$ is indeed a strictly convex function. The following proposition shows that the Bregman divergence induced by the original and the alternate convex function are related by a temperature shift:

**Proposition 3.** *For the Bregman divergence induced by the convex function $\widetilde{F}_t$, we have*

$$\forall \boldsymbol{y}, \hat{\boldsymbol{y}} \in \mathbb{R}_+^k: \quad \Delta_{\widetilde{F}_t}(\boldsymbol{y},\hat{\boldsymbol{y}}) \;=\; \frac{1}{t}\sum_i(\log_t \hat{y}_i - \log_t y_i + (y_i - \hat{y}_i)\hat{y}_i^{-t}) \;=\; \Delta_{F_{t+1}}(\boldsymbol{y},\hat{\boldsymbol{y}}).$$

The $\widetilde{F}_t$ function is also related to the negative Tsallis entropy over the probability measures $\boldsymbol{y} \in \Delta_+^k$ defined as

$$-H_t^{\mathrm{Tsallis}}(\boldsymbol{y}) = \frac{1}{1-t}\Big(1 - \sum_i y_i^t\Big) = -\sum_i y_i \log_t \frac{1}{y_i}.$$

Note that $(-H_t^{\mathrm{Tsallis}} - (1-t)\widetilde{F}_{1-t})$ is an affine function. Thus, the Bregman Divergence induced by $\widetilde{F}_t$, and the one induced by $-H_t^{\mathrm{Tsallis}}$ are also equivalent up to a scaling and a temperature shift. Thus, both functions $F_t$ and $\widetilde{F}_t$ can be viewed as some generalized negative entropy functions. Note that the Bregman divergence induced by $-H_t^{\mathrm{Tsallis}}$ is different from the Tsallis divergence over the simplex, defined as

$$\Delta_t^{\mathrm{Tsallis}}(\boldsymbol{y},\hat{\boldsymbol{y}}) = -\sum_i y_i \log_t \frac{\hat{y}_i}{y_i} = \sum_i y_i^t(\log_t y_i - \hat{y}_i).$$

# D  Convexity of the Tempered Matching Loss

The convexity of the loss function $\Delta_{F_t}\big(\boldsymbol{y}, \exp_t(\hat{\boldsymbol{a}} - \lambda_t(\hat{\boldsymbol{a}}))\big)$ with $t \geqslant 1$ for $\hat{\boldsymbol{a}} \in \mathbb{R}^k$ immediately follows from the definition of the matching loss. A more subtle case occurs when $0 \leqslant t < 1$. Note that the range of the combined function $\log_t \circ \exp_t$ does not cover all $\mathbb{R}^k$ as the $\log_t$ function is bounded from below by $-\frac{1}{1-t}$. Therefore, $\mathbf{range}(\log_t \circ \exp_t) = \{\boldsymbol{a}' \in \mathbb{R}^k \mid -\frac{1}{1-t} \leqslant \boldsymbol{a}'\}$.

**Remark 1.** *The normalization function $\lambda_t(\boldsymbol{a})$ satisfies:* $\lambda_t(\boldsymbol{a} + b\,\mathbf{1}) = \lambda_t(\boldsymbol{a}) + b$ *for* $b \in \mathbb{R}$.

*Proof.* Note that

$$\sum_i \exp_t((a_i + b) - \lambda_t(\boldsymbol{a} + b\,\mathbf{1})) = \sum_i \exp_t\big(a_i - \underbrace{(\lambda_t(\boldsymbol{a}+b\,\mathbf{1})-b)}_{=\lambda_t(\boldsymbol{a})}\big) = 1 \quad \text{for} \quad \forall \boldsymbol{a} \in \mathbb{R}^k.$$

345 The claim follows immediately. □

346 **Proposition 4.** *The loss function* $\Delta_{F_t}\big(\boldsymbol{y}, \exp_t(\hat{\boldsymbol{a}} - \lambda_t(\hat{\boldsymbol{a}}))\big)$ *for* $0 \leqslant t < 1$ *is convex for*

$$\hat{\boldsymbol{a}} \in \{\boldsymbol{a}' + \mathbb{R}\,\boldsymbol{1} \mid -\frac{1}{1-t} \leqslant \boldsymbol{a}'\}\,.$$

347 *Proof.* Using the definition of the dual function $\check{F}^*$ and its derivative $\check{f}^*$, we can write

$$
\begin{aligned}
\Delta_{F_t}(\boldsymbol{y}, \hat{\boldsymbol{y}}) &= F_t(\boldsymbol{y}) - F_t(\hat{\boldsymbol{y}}) - (\boldsymbol{y} - \hat{\boldsymbol{y}}) \cdot f_t(\hat{\boldsymbol{y}}) && \big(\hat{\boldsymbol{y}} = \exp_t(\hat{\boldsymbol{a}} - \lambda_t(\hat{\boldsymbol{a}})\,\boldsymbol{1})\big) \\
&= F_t(\boldsymbol{y}) - F_t(\hat{\boldsymbol{y}}) - (\boldsymbol{y} - \hat{\boldsymbol{y}}) \cdot \log_t \circ \exp_t(\hat{\boldsymbol{a}} - \lambda\boldsymbol{1}) \\
&= F_t(\boldsymbol{y}) - F_t(\hat{\boldsymbol{y}}) - (\boldsymbol{y} - \hat{\boldsymbol{y}}) \cdot (\hat{\boldsymbol{a}} - \lambda_t(\hat{\boldsymbol{a}})\,\boldsymbol{1}) && \big((\boldsymbol{y} - \hat{\boldsymbol{y}}) \cdot \boldsymbol{1} = 1 - 1 = 0\big) \\
&= \underbrace{F_t(\boldsymbol{y}) - \boldsymbol{y} \cdot (\check{f}_t^*)^{-1}(\boldsymbol{y})}_{-\check{F}_t^*((\check{f}_t^*)^{-1}(\boldsymbol{y}))} + \boldsymbol{y} \cdot (\check{f}_t^*)^{-1}(\boldsymbol{y}) \underbrace{-F_t(\hat{\boldsymbol{y}}) + \hat{\boldsymbol{y}} \cdot \hat{\boldsymbol{a}}}_{\check{F}_t^*(\hat{\boldsymbol{a}})} - \boldsymbol{y} \cdot \hat{\boldsymbol{a}} \\
&= \check{F}_t^*(\hat{\boldsymbol{a}}) - \check{F}_t^*((\check{f}_t^*)^{-1}(\boldsymbol{y})) - (\hat{\boldsymbol{a}} - (\check{f}_t^*)^{-1}(\boldsymbol{y})) \cdot \boldsymbol{y} \\
&= \Delta_{\check{F}_t^*}(\hat{\boldsymbol{a}}, (\check{f}_t^*)^{-1}(\boldsymbol{y}))\,.
\end{aligned}
$$

348 Note that the transition from the second line to the third line requires that the assumption $-\frac{1}{1-t} \leqslant \hat{\boldsymbol{a}}$
349 holds. The dual function $\check{F}_t^*$ satisfies

$$\check{F}_t^*(\boldsymbol{a} + b\,\boldsymbol{1}) = \lambda_t(\boldsymbol{a} + b\,\boldsymbol{1}) + \frac{1}{2-t} \sum_i \exp_t\big((a_i + b) - \lambda_t(\boldsymbol{a} + b\,\boldsymbol{1})\big)^{2-t} = \check{F}_t^*(\boldsymbol{a}) + b\,.$$

350 Additionally,

$$\Delta_{\check{F}_t^*}(\hat{\boldsymbol{a}} + b\,\boldsymbol{1}, (\check{f}_t^*)^{-1}(\boldsymbol{y})) = \check{F}_t^*(\hat{\boldsymbol{a}} + b\,\boldsymbol{1}) - \check{F}_t^*((\check{f}_t^*)^{-1}(\boldsymbol{y})) - (\hat{\boldsymbol{a}} + b\,\boldsymbol{1} - (\check{f}_t^*)^{-1}(\boldsymbol{y})) \cdot \boldsymbol{y} = \Delta_{\check{F}_t^*}(\hat{\boldsymbol{a}}, (\check{f}_t^*)^{-1}(\boldsymbol{y}))\,.$$

351 The claim follows by considering the range of $\log_t \circ \exp_t$ and the invariance of the Bregman divergence
352 induced by $\check{F}_t^*$ along $\mathbb{R}\,\boldsymbol{1}$. □

# E  Derivatives of Lagrangian and the Bi-tempered Matching Loss

354 The gradient of $\lambda_t(\boldsymbol{a})$ w.r.t. $\boldsymbol{a}$ can be calculated by taking the partial derivative of both sides of the
355 equality $1 = \sum_j \exp_t(a_j - \lambda_t(\boldsymbol{a}))$ w.r.t. $a_i$:

$$
\begin{aligned}
0 &= \sum_j \exp_t(a_j - \lambda_t(\boldsymbol{a}))^t \Big(\delta_{ij} - \frac{\partial \lambda_t(\boldsymbol{a})}{\partial a_i}\Big) \\
&= \exp_t\big(a_i - \lambda_t(\boldsymbol{a})\big)^t - \frac{\partial \lambda_t(\boldsymbol{a})}{\partial a_i} \sum_j \exp_t(a_j - \lambda_t(\boldsymbol{a}))^t, \quad \text{where } \delta_{ii} = 1 \text{ and } \delta_{ij} = 0 \text{ for } i \neq j. \quad (11)
\end{aligned}
$$

356 Therefore $\frac{\partial \lambda_t(\boldsymbol{a})}{\partial a_i} = \frac{\exp_t\big(a_i - \lambda_t(\boldsymbol{a})\big)^t}{Z_t}$, where $Z_t = \sum_j \exp_t(a_j - \lambda_t(\boldsymbol{a}))^t$. We conclude that $\frac{\partial \lambda_t(\boldsymbol{a})}{\partial a_i}$ is the
357 "*t*-escort distribution" of the distribution $\frac{\exp(a_i - \lambda_t(\boldsymbol{a}))}{Z_1}$.

358 Similarly, the second derivative of $\lambda_t(\boldsymbol{a})$ can be calculated by repeating the derivation on (11):

$$\frac{\partial^2 \lambda_t(\boldsymbol{a})}{\partial a_i \partial a_j} = \frac{1}{Z_t} \sum_{j'} t \exp_t\big(a_{j'} - \lambda_t(\boldsymbol{a})\big)^{2t-1} \Big(\delta_{ij'} - \frac{\partial \lambda_t(\boldsymbol{a})}{\partial a_i}\Big)\Big(\delta_{jj'} - \frac{\partial \lambda_t(\boldsymbol{a})}{\partial a_j}\Big)\,.$$

359 Although not immediately obvious from the second derivative, it is easy to show that $\lambda_t(\boldsymbol{a})$ is in
360 fact a convex $\boldsymbol{a}$. Also the derivative of the loss $L_{t_1}^{t_2}(\hat{\boldsymbol{a}}|\boldsymbol{y})$ w.r.t. $\hat{a}_i$ (expressed in terms of $\boldsymbol{y}$ and
361 $\hat{\boldsymbol{y}} = \exp_{t_2}(\hat{\boldsymbol{a}} - \lambda_{t_2}(\hat{\boldsymbol{a}}))$) becomes

$$
\begin{aligned}
\frac{\partial L_{t_1}^{t_2}}{\partial \hat{a}_i} &= \sum_j \frac{\partial}{\partial \hat{y}_j}\Big(y_j \log_{t_1} y_j - y_j \log_{t_1} \hat{y}_j - \frac{1}{2 - t_1} y_j^{2-t_1} + \frac{1}{2 - t_1} \hat{y}_j^{2-t_1}\Big)\frac{\partial \hat{y}_j}{\partial \hat{a}_i} \\
&= \sum_j (\hat{y}_j - y_j)\, \hat{y}_i^{t_2 - t_1} \Big(\delta_{ij} - \frac{\hat{y}_j^{t_2}}{\sum_{j'} \hat{y}_{j'}^{t_2}}\Big)\,.
\end{aligned}
$$

## F  Proof of Bayes-risk Consistency

The conditional risk of the multiclass loss $\boldsymbol{l}(\hat{\boldsymbol{a}})$ with $l_i := \ell(\hat{\boldsymbol{a}}|y = i)$, $i \in [k]$ is defined as

$$R(\boldsymbol{\eta}, \boldsymbol{l}(\hat{\boldsymbol{a}})) = \sum_i \eta_i \, l_i \,,$$

where $\eta_i := P_{\mathrm{UK}}(y = i | \boldsymbol{x})$.

**Definition 4** (Bayes-risk Consistency). A Bayes-risk consistent loss for multiclass classification is the class of loss functions $\ell$ for which $\hat{\boldsymbol{a}}^\star$, the minimizer of $R(\boldsymbol{\eta}, \boldsymbol{l}(\hat{\boldsymbol{a}}))$, satisfies

$$\arg\min_i \ell(\hat{\boldsymbol{a}}^\star | y = i) \subseteq \mathrm{argmax}_i \, \eta_i \,.$$

*Proof of Proposition 2.* For the bi-tempered loss, we have

$$l_i = -\log_{t_1} \exp_{t_2}(\hat{a}_i - \lambda_{t_2}(\hat{\boldsymbol{a}})) + \frac{1}{2 - t_1} \sum_j \exp_{t_2}(\hat{a}_j - \lambda_{t_2}(\hat{\boldsymbol{a}}))^{2 - t_1} \,.$$

Note that the second term is repeated for all classes $i \in [k]$. Also,

$$R(\boldsymbol{\eta}, \boldsymbol{l}(\hat{\boldsymbol{a}})) = -\sum_i \eta_i \, \log_{t_1} \exp_{t_2}(\hat{a}_i - \lambda_{t_2}(\hat{\boldsymbol{a}})) + \frac{1}{2 - t_1} \sum_i \exp_{t_2}(\hat{a}_i - \lambda_{t_2}(\hat{\boldsymbol{a}}))^{2 - t_1} \,.$$

The minimizer of $R(\boldsymbol{\eta}, \boldsymbol{l}(\hat{\boldsymbol{a}}))$ satisfies

$$\eta_i = \exp_{t_2}(\hat{a}_i^\star - \lambda_{t_2}(\hat{\boldsymbol{a}}^\star)) \,.$$

Since $-\log_{t_1}$ is a monotonically decreasing function for $0 \leqslant t_1 < 1$, we have

$$\arg\min_i \ell(\hat{\boldsymbol{a}}^\star | y = i) = \arg\min_i -\log_{t_1} \exp_{t_2}(\hat{a}_i^\star - \lambda_{t_2}(\hat{\boldsymbol{a}}^\star)) = \arg\max_i \hat{a}_i^\star \subseteq \arg\max_i \eta_i \,.$$

$\square$