[Reviews · NeurIPS 2019]

Reviewer 1



The proposed loss functions seem novel and theoretical analysis are well presented to support their validation. These Bi-Tempered Logistic loss functions are variants of existing ones. They are derived by introducing a temperature into the exponential function and also by replacing the softmax with a high temperature generalization. Besides all these well presented derivation, I’d like to see some statistical properties of these functions, and how they are compared to the existing ones. The authors claim the proposed loss functions are robust to noise and outliers. The authors are encouraged to present more theoretical & emperical analysis on this part. This new loss function is not convex. Although the conventional logistic loss is not convex with respect to some parameters if neural network is used, its convexity still enables researchers to theoretically analyze the performance of the learning algorithm. If this new non-convex function is used, is the analysis still possible?

Reviewer 2



The authors propose a loss which is controlled by two temperature parameters of generalized logarithm (exponential), one of which comes from the probability model and the other comes from Bregman (beta) divergence. The proposed loss shows robust nature to noises as expected and confirmed by simple numerical experiments with good visualization. The idea looks similar with ref[2] using Bregman divergence instead of Tsallis, therefore the proposal is not surprising. In statistics, it is well known Bregman divergences (not only beta-divergence) leads consistent and robust estimators, and heavy-tailed distributions (not only Bregman-dual link function) are insensitive to outliers, so robustness of the proposed loss looks natural outcome. The former facts are intensively investigated by Shinto Eguchi, Frank Nielsen and their collaborators. The latter facts are found in, for example, a famous book by Huber (1981). Overall organization of the paper is well considered including a good introduction of Bregman divergence, and theoretical discussion of the proposed loss is clear enough, so quality and clarity of the paper is very high. The presented idea is nice, but significance to the community is unclear.

Reviewer 3



The paper is well-motivated and introduce a novel tunable class of losses for (DNN) classification by replacing the usual logistic loss. The experiments demonstrate the gain obtained by using this biparametric logistic loss and improve AISTATS'19 - mention general deformed logarithm and exponential (integral of a monotonous function) and then introduce its specialization of Eq. 1 Cite the book of Jan Naudts : "Generalised Thermostatistics", Springer. - explain that parameters are called temperature because of their use in thermostatistics [14]. I think the paper (and notably the abstract) will gain in readibility by not mentioning "temperature" of generalized thermostatistics. - cite book of Amari 2016 "Information Geometry and Its Applications", mention conformal flattening of Tsallis relative entropy, and escort distributions - need to state whether domain is open (convex) or not, and whether the Bregman generator of Legendre-type or not - Should better explain "However, the Tsallis based divergences do not result in proper loss functions" (AISTATS 19 paper) Minor typos: - Kullback Leibler divergence -> Kullback-Leibler divergence - typo 106 -> Kullback-Leibler (KL) divergence

[Author Response · NeurIPS 2019]

We thank all reviewers for their efforts in reviewing our paper, and for the helpful comments and suggestions.

**Reviewer 2:**

- *"Id like to see some statistical properties of these new loss functions, and how they are compared to the existing ones"*:
  We show that our Bregman-based approach leads to proper bi-tempered losses, whereas the previous Tsallis-based losses are not proper.

- *"The authors are encouraged to present more theoretical & empirical analysis on the robustness of these loss functions. . . If this new non-convex function is used, can we still theoretically analyze the performance of the learning algorithm?"*:
  We provide strong empirical evidence of the efficacy of our method, i.e. for dropping the convexity in the activations of the last layer. Proving theoretical convergence under our non-convex losses for a multi-layer neural net is formidable and beyond the scope of the paper.
  However a plausible next goal would be to prove theoretical bounds for a single neuron using our non-convex tunable loss. We leave this up to future work.

**Reviewer 3:** We will add the additional references you suggested. Thanks!

- *"There are many possibilities for combining heavy-tailed distributions and robust/consistent divergence. It is nice to discuss the advantage of the proposed bi-tempered loss: computational cost? mathematical beauty?"*:
  Our method leads to proper losses which is important for many applications that require a probability distribution as output. The computational cost is only negligibly larger than the cost of logistic regression. The foundation of our method is based on generalizations of the exponential family of distributions. The crux of our method is the fact that even in the "many class" case we only have two additional parameters to tune.

**Reviewer 4:** Thanks for your excellent feedback. We will expand on the relationship to the additional references and more clearly contrast our losses with the previous Tsallis-based versions.

[Meta-Review · NeurIPS 2019]

Congratulations to the authors for a very nice, slick paper. It is strongly encouraged to address the few concerns and comments of the reviewers in its final version, in particular R#3, 4.